



**Exploring the drivers of the elevated ozone production in Beijing in**
**summertime during 2005-2016**
Wenjie Wang[1], David D. Parrish[2], Xin Li[1,3,4] *, Min Shao[2,1], Ying Liu[1], Sihua Lu[1], Min
Hu[1], Yusheng Wu[1,#], Limin Zeng[1], Yuanhang Zhang[1]
[1] State Key Joint Laboratory of Environmental Simulation and Pollution Control,
College of Environmental Sciences and Engineering, Peking University, Beijing,
China
[2] Institute for Environmental and Climate Research, Jinan University,
Guangzhou 511443, China
[3] International Joint Laboratory for Regional Pollution Control, Ministry of
Education, Beijing, 100816, China
[4] Collaborative Innovation Centre of Atmospheric Environment and Equipment
Technology, Nanjing University of Information Science & Technology, Nanjing,
210044, China
[#] now at Department of Physics, University of Helsinki, Helsinki, Finland
* Corresponding author.
Address: College of Environmental Sciences and Engineering, Peking
University, Beijing 100871, China
Phone: 86-10-62757973
Email: li_xin@pku.edu.cn





**Abstract**
In the past decade, average $PM_{2.5}$ concentrations decreased rapidly under the
strong pollution control measures in major cities in China; however, ozone ($O_3$)
pollution emerged as a significant problem. Here we examine a unique (for China) 12-
year data set of ground-level $O_3$ and precursor concentrations collected at an urban site
in Beijing (PKUERS), where the maximum daily 8 h average (MDA8) $O_3$ concentration
and daytime Ox ($O_3 + NO_2$) concentration in August increased by $2.3 \pm 1.2$ ppbv ($+3.3$
$\pm 1.8\%$) yr$^{-1}$ and $1.4 \pm 0.6$ ($+1.9 \pm 0.8\%$) yr$^{-1}$ respectively from 2006 to 2016. In contrast,
daytime concentrations of nitrogen oxides (NOx) and the OH reactivity of volatile
organic compounds (VOCs) both decreased significantly. Over this same time, the
decrease of particulate matter, and thus the aerosol optical depth, led to enhanced solar
radiation and photolysis frequencies, with near-surface j($NO_2$) increasing at a rate of
$3.6 \pm 0.8\%$ yr$^{-1}$. We use an observation based box model to analyze the combined effect
of solar radiation and ozone precursor changes on ozone production rate, P($O_3$). The
results indicate that the ratio of the rates of decrease of VOCs and NOx (about 1.1) is
inefficient in reducing ozone production in Beijing. P($O_3$) increased during the decade
due to more rapid atmospheric oxidation caused to a large extent by the decrease of
particulate matter. This elevated ozone production was driven primarily by increased
actinic flux due to $PM_{2.5}$ decrease and to a lesser extent by reduced heterogeneous
uptake of $HO_2$. Therefore, the influence of $PM_{2.5}$ on actinic flux and thus on the rate of
oxidation of VOCs and NOx to ozone and to secondary aerosol (i.e., the major
contributor to $PM_{2.5}$) is important for determining the atmospheric effects of controlling
the emissions of the common precursors of $PM_{2.5}$ and ozone when attempting to control
these two important air pollutants.





## 1 Introduction

Tropospheric ozone ($O_3$) plays a key role in the oxidizing capacity of the atmosphere and affects the global climate; high concentrations of ground-level ozone are harmful to human health and ecosystems (Monks et al., 2015;Fiore et al., 2009). Ozone is produced rapidly in polluted air by photochemical oxidation of volatile organic compounds (VOCs) in the presence of nitrogen oxides ($NOx \equiv NO + NO_2$) (Atkinson, 2000). In recent years, China has undergone rapid economic development, resulting in higher demand for energy, and greater usage of fossil fuels. As a result, high emissions to the atmosphere produce heavy pollution in eastern China, which now suffers from severe ozone pollution, especially in urban areas, where the daily maximum 8 h average (MDA8) ozone level often exceeds the standard of 80 ppb (Jinfeng et al., 2014;Wang et al., 2011;Zhang et al., 2014;Lu et al., 2018;Li et al., 2019). A recent study reported that the national warm-season (April−September) fourth highest MDA8 ozone level (86.0 ppb) and the number of days with MDA8 values of > 70 ppb was much higher than regional averages in Japan, South Korea, Europe, or the United States (Lu et al., 2018). Satellite observations found that regional ozone concentrations in eastern China increased by 7% between 2005 and 2010 (Verstraeten et al., 2015). From 2013 to 2017, the $O_3$ concentrations in 74 cities as a whole showed an upward trend with Beijing-Tianjin-Hebei region being the most serious (Li et al., 2019;Lu et al., 2018). Better understanding of the causes of elevated ozone in China is important for developing effective emission control strategies to reduce the ozone pollution problem.

Aerosols impact ozone production primarily in two ways: alteration of photolysis rates by aerosol radiative influence and heterogeneous reactions occurring on the aerosol surface. The reduction of photolysis frequencies by the extinction effect of aerosol and thus its influence on ozone production has been explored in the past (Dickerson et al., 1997;Castro et al., 2001;Real and Sartelet, 2011;Gerasopoulos et al., 2012;Wang et al., 2019). Absorbing aerosols reduce photolysis frequencies





throughout the boundary layer, and as a result decrease near-surface photochemical
ozone production (de Miranda et al., 2005;Jacobson, 1998;Wendisch et al., 1996;Raga
et al., 2001). Conversely, scattering aerosols in the boundary layer increase photolysis
frequencies throughout the troposphere, and thereby increase ozone production aloft
(Jacobson, 1998;Tian et al., 2019;Dickerson et al., 1997). The importance of aerosol
heterogeneous reactions in ozone photochemistry in China has been previously
investigated in model studies (Lou et al., 2014;Li et al., 2018;Xu et al., 2012;Li et al.,
2019). The effects of $NO_2$, $NO_3$, and $N_2O_5$ heterogeneous reactions showed opposite
$O_3$ concentration changes in VOC-limited and NOx-limited regions. In a VOC-limited
region, $NO_2$, $NO_3$, and $N_2O_5$ heterogeneous reactions lead to ozone concentration
increases (Lou et al., 2014;Xu et al., 2012). The heterogeneous reaction of $HO_2$
decreases ozone production in both VOC-limited and NOx-limited regions by
decreasing the reaction rate of $HO_2$ with NO (Lou et al., 2014;Li et al., 2019).
In the past decade, Eastern China has experienced severe fine particulate matter
($PM_{2.5}$) pollution in winter (Zhang et al., 2016), and this issue has been the main focus
of the government's air pollution control strategy. These stringent emission control
measures have significantly decreased the concentrations of particulate matter in many
Chinese cities. During 2008-2013, ground-level $PM_{2.5}$ estimated from satellite-
retrieved aerosol optical depth (AOD) in China declined at a rate of 0.46 µg m$^{-3}$ year$^{-1}$
(Ma et al., 2016b). Another study indicated that the annual average concentration of
$PM_{2.5}$ in Beijing decreased by 1.5µg m$^{-3}$ year$^{-1}$ and 27% in total from 2000 to 2015
under the implementation of 16 phases' air pollution control measures (Lang et al.,
2017). Hu et al (2017) reported that $PM_{2.5}$ in Beijing declined significantly from 2006
to 2016, and meanwhile solar radiation increased (Hu et al., 2017). However, despite
the reduction in emissions of particulate matter (PM) and ozone precursors, ozone
concentrations increased, even while PM concentrations decreased.
In Beijing, the second largest city in China, with rapid economic development and
urbanization in recent years, ozone pollution is one of the worst among China's cities.
Thus, Beijing is a representative city in which to study urban ozone pollution in China.



Despite extensive study of the relationship between ozone and its precursors in Beijing and other mega cities in China (Zhang et al., 2014;Chou et al., 2011;Lu et al., 2019;Liu et al., 2012), there remains a lack of understanding of the cause of the long-term ozone increase that accompanies reductions in precursor emissions. In this study, we utilize measurements from a representative urban site in Beijing to explore how the variations in solar radiation and heterogeneous reactions influence the trend of ozone and the coupling effect of aerosol and ozone precursor changes on ozone production. Our overall goal is to determine the extent to which increasing actinic flux caused by the decline in PM contributed to the observed increase in ozone concentrations. This research provides a clearer understanding of how efforts to reduce PM concentrations affect ozone concentrations, and thus informs air quality improvement efforts in China's urban areas.

## 2 Materials and methods

### 2.1 Measurements of air pollutants, photolysis frequencies and aerosol surface concentration

Ambient air pollutants and photolysis frequencies were measured at an urban site in Beijing in August between 2005 and 2016. The site (39.99° N, 116.31°E) was located on the roof of a six story building (~20m above the ground level) on the campus of Peking University (PKUERS) near the 4th Ring Road with high density of traffic, but without obvious industrial or agricultural sources (Wehner et al., 2008). Temporal trends of air pollutants and composition of VOCs are thought to be representative for the whole of Beijing (Wang et al., 2010;Xu et al., 2011;Zhang et al., 2012). Measured parameters include $O_3$, NOx, CO, $SO_2$, C2 - C10 VOCs, photolysis frequencies and aerosol surface concentration. The measurement techniques are included in the Table 1.

During 2006 and 2008, ambient levels of VOCs were measured using an online GC-FID system built by the Research Center for Environmental Changes (RCEC; Taiwan). A detailed description of this system and QA/QC procedures can be found in





Wang et al. (Wang et al., 2004). During August 2007 and 2009, ambient VOCs were
measured using a commercial GC-FID/PID system (Syntech Spectra GC955 series
600/800 analyzer) (Xie et al., 2008;Zhang et al., 2014). From 2010 to 2016, VOCs were
measured using a cryogen-free online GC-MS/FID system developed by Peking
University. A detailed description of this system and QA/QC procedures can be found
in Yuan et al. and Wang et al. (Yuan et al., 2012;Wang et al., 2014). Formaldehyde
(HCHO) concentrations were measured by a Hantzsch fluorimetry.

Photolysis frequencies (including $j(O^1D)$, $j(NO_2)$, $j(HONO)$, $j(HCHO)\_M$,

$j(HCHO)\_R$, $j(H_2O_2)$) were calculated from solar actinic flux spectra measured by a
spectroradiometer as described by Bohn et al. (Bohn et al., 2008). The particle number
size distributions were measured by a system consisting of a Nano-SMPS (TSI
DMA3085 + CPC3776) and a SMPS (TSI DMA3081 + CPC3775). Aerosol surface
concentration (Sa) during 2006-2016 was calculated from the measured particle number
size distributions between 3 nm and 700 nm by assuming the particles are spherical in
shape.

**2.2 Estimate of photolysis frequencies**

Photolysis frequencies were measured in August 2011-2014 and 2016. The

Tropospheric Ultraviolet and Visible (TUV) radiation model (version 5.3) was used to
calculate photolysis frequencies in August over the entire 2006-2016 period under
clear-sky conditions. TUV uses the discrete-ordinate algorithm (DISORT) with four
streams and calculates the actinic flux spectra with a wavelength range of 280 – 420 nm
in 1nm steps and resolution. We used observed aerosol optical properties including
AOD, single scattering albedo (SSA) and Ångström exponent (AE), total ozone column
to constrain the TUV model (Madronich, 1993). The calculated values agree well with
measured results as shown in Figure 1 indicating that the TUV model accurately
calculated the photolysis frequencies. Data of photolysis frequencies under cloudless
conditions were selected according to the presence of AOD data since AOD
measurements were not possible under cloudy conditions.




## 2.3 Measurements of aerosol optical properties

Aerosol optical properties were measured with a CIMEL Sun photometer
(AERONET level 1.5 and level 2.0 data collection, http://aeronet.gsfc.nasa.gov/) at the
Beijing-CAMS site (39.933°N, 116.317°E) and at the Beijing site (39.977N,116.381E).
The instrumentation, data acquisition, retrieval algorithms and calibration procedure,
which conform to the standards of the AERONET global network, are described in
detail by Fotiadi et al. (Fotiadi et al., 2006). The solar extinction measurements taken
every 3 minutes within the spectral range 340 – 1020 nm were used to compute AOD
at 340, 380, 440, 500, 675, 870, 970 and 1020 nm. The overall uncertainty in AOD data
under cloud-free conditions was 0.02 at a wavelength of 440 nm (Dubovik and King,
2000). In this study, AOD at the wavelength of 380nm was chosen for analysis. This
wavelength was selected as it is more representative of $j(NO_2)$. In addition to AOD, that
network also provided single scattering albedo (SSA) and Ångström exponent (AE)
data.
Cloud optical thickness (COT) was acquired from Aura satellite measurements
with a time resolution of 24 hours. Total ozone column was obtained by OMI (Ozone
Monitoring Instrument), using overpass data.

## 2.4 Chemical box model

Ozone production rate, $P(O_3)$, is calculated by a chemical box model. This model
is based on the compact Regional Atmospheric Chemical Mechanism version 2
(RACM) described by Goliff et al. (Goliff et al., 2013), which includes 17 stable
inorganic species, 4 inorganic intermediates, 55 stable organic compounds and 43
intermediate organic compounds. Compounds that are not explicitly treated in the
RACM are lumped into species with similar functional groups. The isoprene
mechanism includes a more detailed mechanism based on the Leuven Isoprene
Mechanism (LIM) proposed by Peeters et al. (Peeters et al., 2009). A detailed



description of this model can be found in Tan et al. (Tan et al., 2017).

In this study, the model was constrained by measured hourly average CO, $NO_2$,

$O_3$, $SO_2$, NMHCs (56 species), HCHO, photolysis frequencies, temperature, pressure,
and relative humidity. HONO was not measured. HONO concentrations are generally
underestimated by the gas phase reaction source of HONO (OH + NO → HONO) in
urban areas due to the emission of HONO and the heterogeneous reaction of NOx at
surfaces to form HONO, both of which are related to NOx concentration. As a result,
the HONO concentration was calculated according to the concentration of $NO_2$ and
the observed ratio of HONO to $NO_2$ at an urban site in Beijing, which had a marked
diurnal cycle (Hendrick et al., 2014). For the model calculation, the ratio of HONO to
$NO_2$ is equal to 0.08 at 6:00 and decreases linearly from 0.08 to 0.01 during 6:00 -
10:00 reflecting increasing photolysis of HONO, and maintains the value of 0.01
during 10:00-18:00. In this study, we focused on daytime $P(O_3)$ (6:00 - 18:00), thus
the nocturnal HONO concentrations were not required.

$RO_2$, $HO_2$, OH were simulated by the box model to calculate the ozone

production rate as shown in Equation E1 and E2 as derived by Mihelcic et al.
(Mihelcic et al., 2003).
$$P\left(O_3\right) = k_{HO_2+NO}\left[HO_2\right]\left[NO\right] + \Sigma\left(k^i_{RO_2+NO}\left[RO^i_2\right]\left[NO\right]\right) - k_{OH+NO_2}\left[OH\right]\left[NO_2\right] - L\left(O_3\right) \quad \text{E1}$$
$$L\left(O_3\right) = \left(\theta\, j\left(O^1D\right) + k_{OH+O_3}\left[OH\right] + k_{HO_2+O_3}\left[HO_2\right] + \Sigma\left(k^j alkene + O_3\left[alkene^j\right]\right)\left[O_3\right]\right) \quad \text{E2}$$
where θ is the fraction of $O^1D$ from ozone photolysis that reacts with water vapor. i and
j represent the number of species of $RO_2$ and alkenes, respectively.

The model runs were performed in a time-dependent mode with two days' spin-

up. A 24 h lifetime was introduced for all simulated species, such as secondary species
and radicals, to approximately simulate dry deposition and other losses of these
species (Lu et al., 2013). This lifetime corresponds to an assumed deposition velocity
of 1.2 cm $s^{-1}$ and a well-mixed boundary layer height of about 1 km. Sensitivity tests
show that this assumed deposition lifetime has a relatively small influence on the
reactivity of modeled oxidation products and ROx radicals.

Aerosols can influence $O_3$ production by heterogeneous reactions such as uptake



of $HO_2$, $NO_2$, $N_2O_5$ and $NO_3$. For these species, the heterogeneous uptake of $HO_2$ is
expected to have the largest effect on rapid ozone production in summertime and VOC-
limited conditions (Li et al., 2019). Thus, the effect of heterogeneous reaction of $HO_2$
on ozone production was simulated in the chemical box model using RH corrected
aerosol surface concentration ($S_{aw}$) and uptake coefficient of $HO_2$. The rate of change
in $HO_2$ due to irreversible uptake is expressed by E3.
$$\frac{dC}{dt} = \frac{\gamma_{HO_2} \times S_{aw} \times v \times C}{4}$$    E3
Where C , v , and $\gamma_{HO_2}$ are the gas phase concentration, mean molecular velocity, and
uptake coefficient, respectively. To derive $S_{aw}$ we used the measured hygroscopic
factor (Liu et al., 2009) and measured RH to correct the measurement-derived $S_a$ to
ambient conditions. In this study, we chose $\gamma_{HO2} = 0.2$ provided by laboratory
measurements of $HO_2$ uptake by aerosol particles collected at two mountain sites in
eastern China (Taketani et al., 2012). The effects of $HO_2$ uptake on $P(O_3)$ in Beijing in
2006 were simulated assuming that the product of $HO_2$ uptake by aerosols is either
$H_2O$ or $H_2O_2$. The results indicate that the two scenarios showed no significant
difference because the recycling of HOx radicals from $H_2O_2$ is inefficient (Li et al.,
2019). In the following simulations in this study, the product of $HO_2$ uptake by
aerosols is taken to be $H_2O$.
**3 Results and discussion**
**3.1 Trend of ozone**
Ozone pollution levels can be characterized by a number of metrics. Table 2 lists
10 ozone metrics and their definition summarized by Lu et al. (Lu et al., 2018). We
classify these indicators into four categories: (1) metrics that characterize general levels
of ozone: median value of hourly ozone concentrations (median), daily maximum 8 h
average ozone concentration (MDA8) and daytime average ozone concentration
(DTAvg); (2) metrics that characterize extreme levels of ozone: daily maximum 1 h



average ozone concentration (MDA1), 98th percentile of hourly ozone concentrations
(Perc98) and 4th highest MDA8 (4MDA8); (3) metrics that characterize ozone
exposure: cumulative hourly ozone concentrations of >40 ppb (AOT40) and sum of
positive differences between MDA8 and a cutoff concentration of 35 ppb (SOMO35);
(4) The metrics that characterize the days when the ozone exceeds the standard: total
number of days with MDA8 values of >70 ppb (NDGT70) and number of days with
the ozone concentration exceeding the Chinese grade II national air quality standard
(Exceedance). Figure 2 presents variations in these four categories of ozone metrics at
PKUERS site during the study periods. The results show that overall all metrics
increased during the 12 year period. However, the percent increase and the correlation
coefficient of each metric are different. The median, DTAvg, and MDA8 indicators,
which characterize the general concentration levels of ozone, had an increase rate of
2.8% ~ 5.7% yr$^{-1}$. The metrics that characterize the extreme concentration levels of
ozone had a slower increase rate of 1.2% ~ 2.7% yr$^{-1}$. Among them, Perc98 had the
smallest rise rate, only 1.2% yr$^{-1}$, and the correlation is not significant ($r^2 = 0.11$). This
suggests that the extreme ozone pollution increased much less significantly. In contrast,
the increase rates of the ozone exposure metrics AOT40 and SOMO35 was are faster,
8.4% yr$^{-1}$ and 8.3% yr$^{-1}$, respectively, than the metrics that characterize ozone
concentrations. The NDGT70 and Exceedance metrics, related to the number of days
of ozone exceeding the standard, showed the fastest increases, 10% yr$^{-1}$ and 9.8% yr$^{-1}$,
respectively. It worth noting that most of metrics decreased significantly from 2014 to
2016 except for Perc98 and 4MDA8.
As shown in Figure 3, from 2005 to 2016 MDA8 $O_3$ concentrations increased at a
rate of $2.3 \pm 1.2$ ppbv ($3.3 \pm 1.8$ %) yr$^{-1}$ ($r^2 = 0.66$) at the PKUERS site, which
corresponds to a total MDA8 ozone increase of 25.3 ppbv. Meanwhile, $O_X$ ($O_3+NO_2$)
concentrations increased at a slower rate of $1.4 \pm 0.6$ ppbv ($1.9 \pm 0.8$ %) yr$^{-1}$, due to the
decrease in NOx concentrations (second graph in Figure 2).
Temperature and wind speed, which can directly influence ozone production and
concentrations, showed no significant trend during 2005-2016 (Figure 4). The average



temperatures in summer were between 26 and 31°C. The temperature in 2005 was the
lowest and in 2007 it was the highest. The average wind speeds were less than 2.5 m s$^-$
$^1$ in all years. The average relative humidity may have decreased slightly ($\sim$ 1.5% yr$^{-1}$).
In summary, we believe that meteorological factors did not play more than a minor role
in the overall Beijing $O_3$ trend. Therefore, our discussion focuses on photochemical
processes.
The ozone concentration observed at a receptor site depends on two contributions:
regional background ozone and local photochemical production. We have no direct
measurements of the long-term trend of regional background ozone in Beijing, but
others have reported measurements of ozone at regional background sites in China. At
a baseline Global Atmospheric Watch (GAW) station in the northeastern Tibetan
Plateau region (Mt Waliguan, 36.28° N, 100.9° E) the average annual daytime ozone
concentration increased at a rate of 0.24 ppb yr$^{-1}$, over the 1994 to 2013 period, but
there was no significant trend in summer (Xu et al., 2018). The measurement at a rural
station in Beijing (116.22° E, 40.29° N, 34 km northwest of the observation site in this
study) showed a decrease of ozone at a rate of -0.47 ppb yr$^{-1}$ over the 2004 to 2015
period (Zheng et al., 2016). The MDA8 ozone concentration at the Shangdianzi site, a
background station in northern China, showed an increasing trend of 1.1 ppb yr$^{-1}$ during
2004-2014 (Ma et al., 2016a). Additionally, there were very small trends of $O_3$
concentrations at the background site (Dongtan) in Shanghai, located in the southern
North China Plain (Gao et al., 2017). Based on these reports of smaller and variable
trends, we assume that the trend in regional background ozone in the North China Plain
made only a minor contribution to the ozone trend observed at the PKUERS site (2.3 $\pm$
1.2 ppbv yr$^{-1}$). We thus surmise that the increase in $O_3$ at the PKUERS site was mainly
due to "local" photochemistry driven by emissions of ozone precursors from downtown
and the surrounding suburban areas of Beijing.
**3.2 Trend of gaseous precursors**
This increase in ozone concentrations is opposite to the decreasing trend of its





precursors, including VOCs, CO and NOx (Figure 5). The overall change of the total
OH loss rate due to VOCs (VOC reactivity) was -0.36 $s^{-1}$ (-6.0%) $yr^{-1}$. For
anthropogenic VOCs, the highest reactivity was generally contributed by alkene
species, with an average value over the eleven years of $2.00 \pm 0.43$ $s^{-1}$, followed by
aromatics and alkanes, with average reactivities of $1.51 \pm 0.74$ $s^{-1}$ and $0.92 \pm 0.60$ $s^{-1}$,
respectively. Thus, the alkenes and aromatics are more important for $O_3$ production
than are alkanes. The trends for alkenes, aromatics, and alkanes were a decrease of
0.14 $s^{-1}$ (7.1%), 0.12 $s^{-1}$ (7.9%), and 0.065$s^{-1}$ (7.0%) $yr^{-1}$, respectively, indicating that
alkenes and aromatics played the dominant role in the reduction of anthropogenic
VOC reactivity. The rate of decrease in VOCs at PKUERS site is similar to that
reported for Los Angeles by Warneke et al. and Pollack et al. (7.3-7.5% $yr^{-1}$ over 50
years) (Warneke et al., 2012;Pollack et al., 2013). The decrease in anthropogenic
VOCs in Los Angeles was predominantly attributed to decreasing emissions from
motor vehicles due to increasingly strict emissions standards. Similarly, a previous
study at the PKUERS site indicated that the decreasing anthropogenic VOC was
mainly attributed to the reduction of gasoline evaporation and vehicular exhaust under
the implementation of stricter emissions standards for new vehicles and specific
control measures for in-use vehicles (Wang et al., 2015a). For naturally emitted
VOCs, mainly isoprene, the OH reactivity had little trend with large fluctuations, as
the emissions of plants vary greatly with temperature and light intensity. Therefore,
the decrease in total VOCs reactivity was dominated by the decrease in anthropogenic
VOCs. Similarly, CO, which is mainly contributed by anthropogenic emissions,
decreased rapidly (9.3% $yr^{-1}$) during 2006–2016.
Daytime concentrations of NOx at the PKUER site also decreased significantly
from 2006 to 2016 (Figure 5), with a slope (excluding 2008, which had a much lower
NOx concentration due to enhanced emission controls implemented during the Olympic
Games) of -1.48 ppbv $yr^{-1}$ ($-5.5\%$ $yr^{-1}$, $r^2 = 0.81$). The decrease in NOx was mainly
due to the reduction in vehicle exhaust and coal combustion (Zhao et al., 2013). The
decrease in NOx was significantly faster than that found in Los Angeles by Pollack et



al. (2.6% $yr^{-1}$ over 50 years) (Pollack et al., 2013). In contrast to Beijing, Los Angeles
$O_3$ concentrations have continuously decreased from 1980 to 2010 (Parrish et al., 2016).
The ratio of the rates of decrease of VOCs and NOx in Los Angeles (2.9) is significantly
greater than unity and larger than that at the PKUER site (1.1), which possibly can be
a contributing cause of the opposite trends of ozone in the two regions. It worth noting
that the precursor concentrations in 2008, the Olympic Games year, were particularly
low, but that ozone was nevertheless on the regression line. The monthly average ratio
of VOC reactivity to NOx concentration in 2008 is 0.28 $s^{-1}$ $ppbv^{-1}$, higher than the
average ratio of VOC reactivity to NOx concentration during 2006-2016 (0.24 $s^{-1}$ $ppbv^{-1}$
). The adverse reduction ratio of VOC to NOx is the main cause of inefficient reduction
in $O_3$ level in 2008, which is consistent with the study of Chou et al. (2011).
Since 2013, under the implementation of the Action Plan on Air Pollution
Prevention and Control (http://www.gov.cn/zwgk/2013-09/12/content_2486773.htm),
more stringent emission control measures were implemented to restrict industrial and
vehicle emission. As a result, there are indications that both VOCs and NOx decreased
faster over the 2013 to 2016 period: 0.81 $s^{-1}$ $yr^{-1}$ (16% $yr^{-1}$, $r^2 = 0.71$) and 1.94 ppbv $yr^{-1}$
(9.3% $yr^{-1}$, $r^2 = 0.78$) for VOC reactivity and NOx, respectively. This could be the
cause of the decline in $O_3$ concentrations from 2014 to 2016.

**3.3 Trend of particulate matter**

From 2009 to 2016, $PM_{2.5}$ concentrations declined rapidly, achieving the air
quality standard of China (35 $\mu g/m^3$) in 2016 (Figure 6). Since 2000, Beijing had
implemented 16 phases' air pollution control measures, mainly including the
controlling of industry, motor vehicle, coal combustion and fugitive dust pollution,
which was effective for the reduction in $PM_{2.5}$ (Lang et al., 2017). Especially the
strengthening of the reduction in coal combustion, which was gradually replaced by
natural gas since 2004, favored improved visibility in Beijing (Zhao et al., 2011).
As shown in Figure 6, from 2006 to 2016 AOD decreased at a rate of 9.3% $yr^{-1}$.
The correlation between AOD and $PM_{2.5}$ can be determined from the observations of


PM$_{2.5}$ and AOD in August during 2009-2016 at the PKUERS site (Figure 7). AOD
and PM$_{2.5}$ are linearly correlated with a correlation coefficient of +0.74. This result
indicates that the decrease in PM$_{2.5}$ was the primary cause of the reduction in AOD. In
addition to PM$_{2.5}$, relative humidity also has an important effect on AOD. The
decrease in relative humidity during 2006-2016 (Figure 4) would reduce the
hygroscopic growth of aerosol, leading to a weakened extinction effect of particulate
matter on solar radiation (Qu et al., 2015). It is worth noting that although PM$_{2.5}$ in
2011 was lower than that in 2010, AOD in 2011 was higher than that in 2010 (Figure
6). For one reason, the relative humidity in 2011 was higher. Additionally, the aerosol
type, atmospheric boundary layer height and the vertical structure of aerosol
distribution also affects the dependence of AOD on PM$_{2.5}$ (Zheng et al., 2017),
probably contributing to the scatter about the AOD versus PM$_{2.5}$ relationship shown in
Figure 7.
Monthly mean AE (380/550 nm) in August showed no overall trend during 2006-
2016 (Figure 8). The monthly AE means were between 0.87 and 1.2, suggesting that
the size-distribution of aerosol was generally stable during this period. Monthly mean
SSA (440 nm) in August showed an upward trend of +0.004 yr$^{-1}$ (+0.45% yr$^{-1}$) during
2006-2016 (Figure 8), indicating the proportion of the light-absorbing component of
aerosols (e.g. black carbon) has decreased, due to the stringent and effective controls
on the burning of biomass/biofuel and coal (Ni et al., 2014;Cheng et al., 2013). This
result is consistent with the studies of Lang et al. and Wang et al., which indicated that
black carbon in China's mega cities has decreased rapidly over the past decade (Wang
et al., 2016;Lang et al., 2017).
**3.4 Trend of photolysis frequencies**
The influence of solar radiation on O$_3$ photochemistry can be described by
actinic flux (or photolysis frequencies). We chose j(NO$_2$) as a representative
photolysis frequency to analyze the trend of actinic flux. Wang et al (2019) studied the
quantitative relationship between j(NO$_2$) and AOD at the PKUERS site, and found



that $j(NO_2)$ and AOD showed a clear nonlinear negative correlation at a given SZA,
with slopes ranging from -1.3 to $-3.2 \times 10^{-3}$ $s^{-1}$ at AOD < 0.7, indicating a significant
extinction effect of AOD on actinic flux near the ground.
The $j(NO_2)$ calculated by the TUV model under clear-sky conditions shows an
upward trend of 3.6% $yr^{-1}$ from 2005 to 2016 and agrees well with the 5 years of
observed values from 2011 to 2016 (Figure 6). According to sensitivity analysis of
TUV, the decrease in AOD plays a dominant role in the $j(NO_2)$ increase, contributing
about 80% of the total. Additionally, the increase in SSA also contributes significantly
to $j(NO_2)$ increase, contributing about 17%.
In addition to aerosol optical properties, the photolysis frequency in the planetary
boundary layer is affected by other factors, including cloud extinction, ground
reflection, absorption by gases such as $O_3$, and Rayleigh scattering by gases. The
ground reflection is relatively stable for different years in the same city with stable
ground covering. The change in Rayleigh scattering of gases and absorption of $NO_2$,
$SO_2$ and HCHO plays a negligible role in the variation in photolysis frequencies
according to sensitivity analysis of TUV model. This is consistent with the results of
Barnard et al. (Barnard et al., 2004). As shown in Figure 9, the total ozone column
fluctuated between 285-307 DU without a significant overall trend. The magnitude of
total ozone column variation (22 DU) can leading $j(O^1D)$ changing of about 10%, but
plays a negligible role in changing other photolysis frequencies according to
sensitivity analysis using the TUV model. The cloud optical thickness (COT) for most
years was relatively stable, ranging from 6 to 8, but in 2005, 2012 and 2015 COT was
significantly larger (Figure 9). As there was no significant trend of COT, we surmised
that the light-extinction effect of clouds did not play a key role in changing photolysis
frequencies.
**3.5 Combined effect of changes in ozone precursors and aerosols on ozone**
**production**
We investigated the overall effect of the changes in VOCs, NOx, photolysis



frequency, and aerosol uptake of $HO_2$ on ozone production rate using the chemical
box model. By testing the response of $P(O_3)$ as calculated from Equation E1 to the
changes of VOCs and NOx concentrations (Figure 10), we concluded that
photochemical environment of the PKUERS site was, on average, in the VOC-limited
regime. This result is consistent with previous studies (Zhang et al., 2014;Chou et al.,
2011). Under this condition, the long-term decrease in VOCs in Beijing has
contributed to a decrease in $P(O_3)$, while the decrease in NOx has tended to increase
$P(O_3)$. As shown in Figure 11, when the increase in photolysis frequencies and aerosol
uptake of $HO_2$ were not included in the calculation, the simulated $P(O_3)$ decreased
slightly at a rate of 1.1% $yr^{-1}$. This indicates that the ratio of the rates of decrease of
VOCs and NOx (about 1.1) is nearly inefficient in reducing ozone production in
Beijing. However, when the increase in photolysis frequencies was included in the
model calculation, the calculated $P(O_3)$ showed an increasing trend of 2.2% $yr^{-1}$. This
result indicates that the increase in photolysis frequencies more than compensated for
the downward trend of $O_3$ production driven by decreased VOCs and NOx, leading to
increasing $O_3$ production through the decade. The photochemical box model
calculations indicate that the increase in photolysis frequencies has two major impacts
on $P(O_3)$ - an increase in primary production of OH through accelerated photolysis of
$O_3$, HONO, HCHO and other carbonyl compounds, and an accelerated radical
recycling of OH as VOCs are oxidized. As particulate matter has decreased and
photolysis frequencies correspondingly have increased, a more rapidly decreasing rate
of the VOC to NOx ratio is required to achieve a significant reduction in $O_3$ in the
future.

When we include heterogeneous uptake of $HO_2$ in the model, the calculated

$P(O_3)$ increases at the faster rate of 2.9% $yr^{-1}$ due to the overall reduced aerosol
surface concentration ($S_a$) and thus the reduced heterogeneous uptake of $HO_2$ (Figure
11). This result indicates that the effect of heterogeneous uptake of $HO_2$ contributed
roughly 0.7% $yr^{-1}$ of ozone increase. Hence, our result indicates that the increase in
photolysis rates due to PM decrease plays a more important role than the decrease in





heterogeneous uptake of $HO_2$ by aerosols in accelerating ozone production in Beijing.
Previous measurements indicate that the uptake coefficient varies widely from 0.003
to 0.5 with a strong dependence on the aerosol concentration of transition metal ions
such as Cu(II) (Zou et al., 2019;Taketani et al., 2008;Lakey et al., 2015;Matthews et
al., 2014;Lakey et al., 2016). This strong dependence on aerosol composition implies
that a single assumed value for $\gamma_{HO2} = 0.2$ has large uncertainty. $\gamma_{HO2} = 0.2$ used in our
simulation is likely an overestimate of the effect of heterogeneous uptake of $HO_2$ on
ozone production rate at PKUERS site.
In summertime PM in the Beijing urban area is mainly formed by the secondary
conversion of gaseous precursors (Han et al., 2015;Guo et al., 2014), indicating that
VOCs and NOx are not only the precursors of ozone, but also the main precursors of
PM in this urban area. In addition, observations in Beijing have shown that the
secondary components of PM, including secondary organic matter, ammonium sulfate
and ammonium nitrate, dominate the light extinction of PM (Han et al., 2014;Han et al.,
2017;Wang et al., 2015b). As a result, reductions of VOCs and NOx are expected to
lead to a decrease in secondary PM formation, and thus to further enhancement in solar
radiation (or actinic flux). Therefore, in order to reduce ozone effectively, the
contribution of VOCs and NOx to secondary PM formation and thus their effect on
solar radiation must be comprehensively considered. However, the summertime
formation of PM is quite complex; the conversion efficiency of gaseous precursors to
aerosols and in turn influence ozone production is a research area that requires further
study.
**3.6 Additional considerations**
One limitation of this study is that the photochemical box model is constrained
by surface observations, and hence may not accurately represent some aspects of the
photochemistry through the full depth of the planetary boundary layer over Beijing.
Here we briefly consider several of these aspects: (1) The treatment of ozone and
VOC and NOx precursor concentrations likely are accurately represented, because





rapid daytime vertical mixing ensures that there is only a small vertical gradient in the
concentrations of these relatively long lived species. (2) In daytime, the HONO
lifetime is so short that it may be largely confined to near the surface, where it has
surface sources (heterogeneous reaction of $H_2O$ and $NO_2$ and emissions on surfaces).
Therefore, the estimated HONO based on near-surface $NO_2$ concentrations may
overestimate average boundary layer HONO concentrations; however in this study the
influence of HONO on the calculation is relatively small, so this is not a large source
of error. (3) The model is constrained by surface measurements of photolysis
frequencies, but these surface measurements do not accurately quantify the actinic
flux throughout the boundary layer. Figure 12 presents the vertical profiles of $j(NO_2)$
simulated by the TUV model for aerosol properties representative of Beijing. A thick
layer of aerosol effectively reduces radiation at the bottom of the layer, but not at the
top, where radiation may be enhanced due to upward scattering from the aerosol
below (Dickerson et al., 1997;Jacobson, 1998). Overall, vertical average $j(NO_2)$
increased by 32% from 2006 to 2016, which is comparable to the surface increase
(36%). These simulations indicate that the increased trend of $j(NO_2)$ derived from
surface observations do approximate the trend through the entire boundary layer.
**4 Conclusion**
During the past decade, China has devoted very substantial resources to
improving the environment. These efforts have improved atmospheric particulate
matter loading, but ambient ozone levels have continued to increase. Based on the
long-term measurements at a representative site in Beijing, we explored the factors
driving the increase in ozone production. Consistent with the implementation of
stringent emission control measures, concentrations of $PM_{2.5}$ and ozone precursors
(VOCs and NOx) decreased rapidly, but in contrast $O_3$ and Ox increased. This
investigation finds that the primary cause of the $O_3$ increase is that decreasing PM
concentrations led to an increase in actinic flux, which in turn increased the





photochemical production of ozone. This result indicates that the influence of aerosol
on ozone production is important for determining the full manifold of atmospheric
effects that result from reducing the emissions of the $O_3$ and PM precursors.



**ACKNOWLEDGEMENTS**
This work was supported by the Major Program of the National Natural Science
Foundation of China  [Grant number 91644222]. We thank Hongbin Chen and
Philippe Goloub for data management of AOD and other aerosol optical properties on
AERONET.











Figure 1. Correlation between Observed and calculated $j(NO_2)$ by TUV model in

Beijing in summer time during 2012 - 2015.




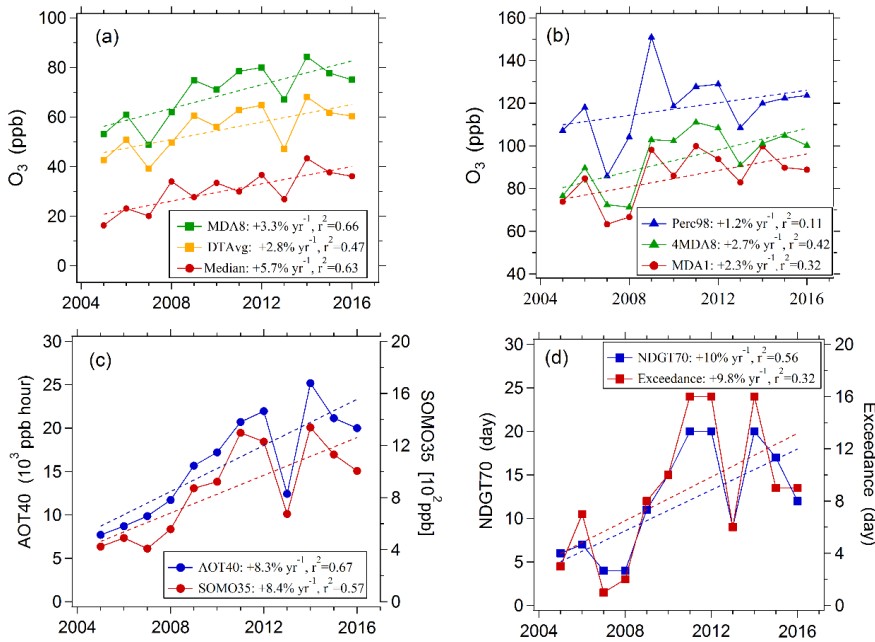

Figure 2. Variations in multiple O₃ metrics at the PKUERS site in Beijing in August
between 2005 and 2016.



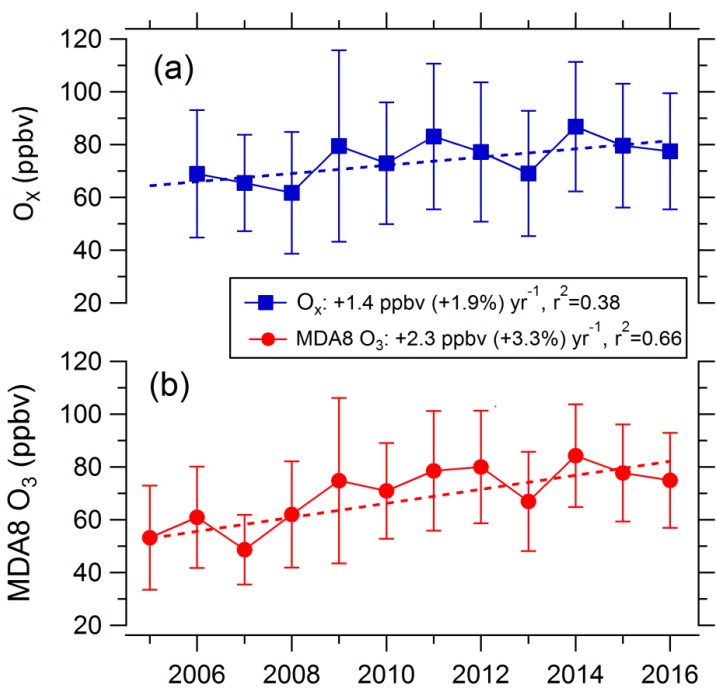


Figure 3. Variations in average MDA8 $O_3$ and daytime (7:00-19:00) average Ox in
Beijing, August between 2005 and 2016.



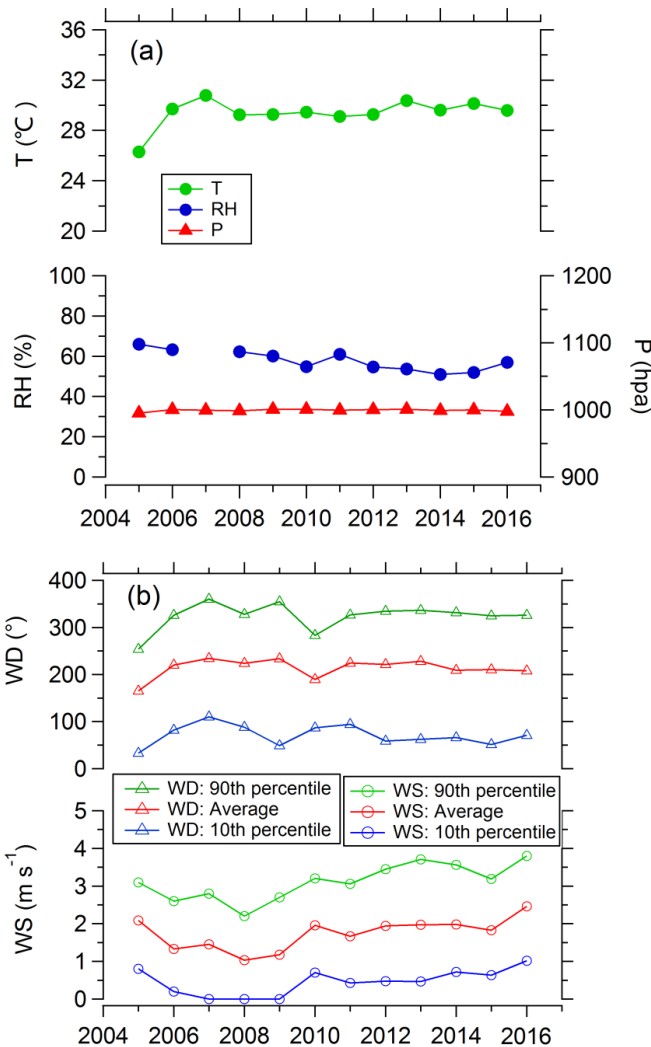

Figure 4. Variations in daytime (7:00-19:00) averages of meteorological conditions including temperature (T), relative humidity (RH), wind direction (WD) and wind speed (WS) in Beijing, August during 2005 - 2016.

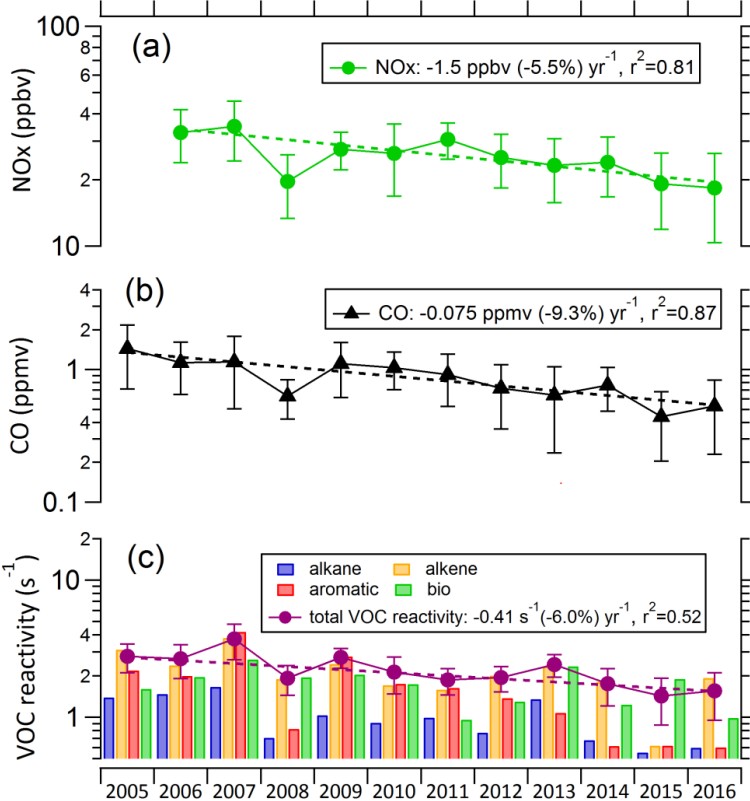

Figure 5. Variations in arithmetic mean MDA8 $O_3$, arithmetic mean of daytime (7:00-

19:00) Ox and geometric mean of daytime NOx, CO and VOCs reactivity in Beijing,

August between 2005 and 2016. VOCs reactivity is depicted by reactivity of each

species (left axis) and total VOC reactivity (right axis). On the y-axes, a linear scale is

used for $O_3$ and Ox, and a log-scale is used for the precursor concentrations (NOx,

CO and VOCs).







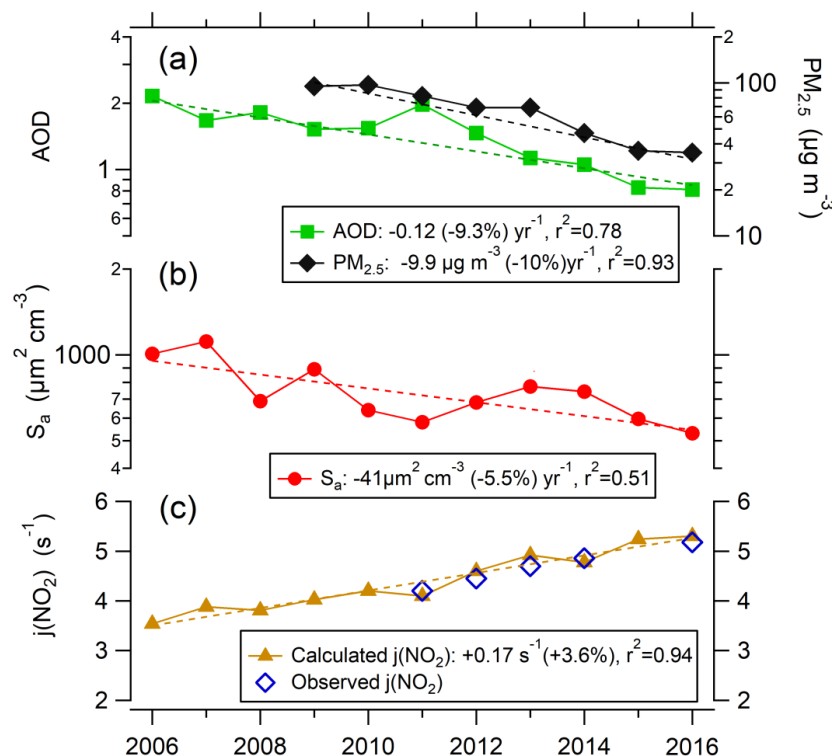


Figure 6. Variations in daytime (7:00-19:00) averages of AOD, $PM_{2.5}$, $S_a$, $j(NO_2)$
Calculated $j(NO_2)$ by TUV in Beijing, August between 2006 and 2016. AOD and
$j(NO_2)$ are both corresponding to cloudless weather. On the y-axes, a log-scale is used
for $PM_{2.5}$, AOD and $S_a$ and a linear scale is used for $j(NO_2)$.








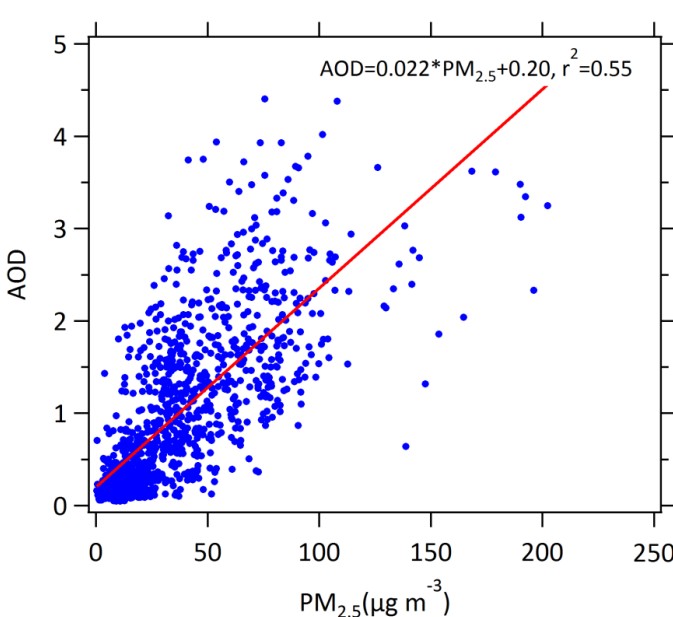


Figure 7. Correlation between AOD and $PM_{2.5}$ in Beijing, summertime during 2009 -

2016.







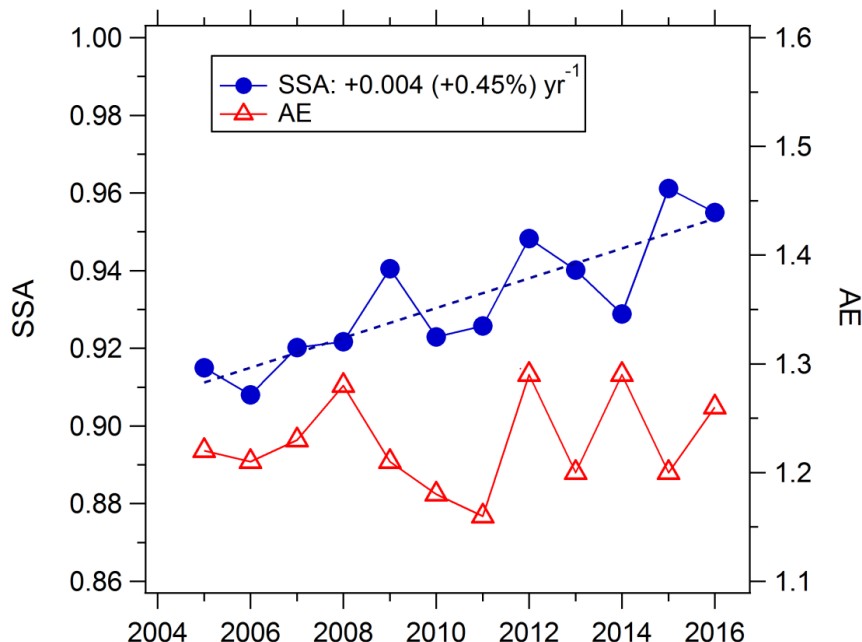


Figure 8. Variation in monthly mean single scattering albedo (SSA) and Ångström
exponent (AE) in Beijing for the month of August during 2005 - 2016.





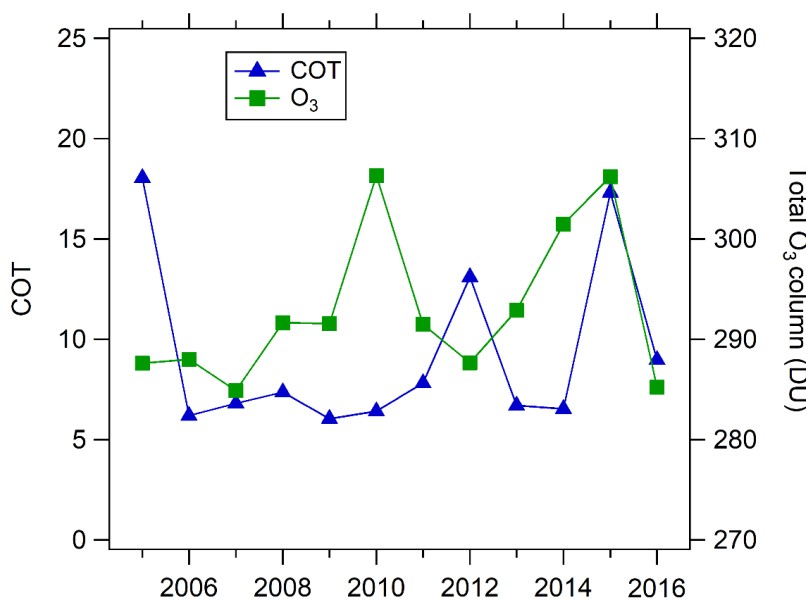


Figure 9. Variations in mean total ozone column and cloud optical thickness (COT) in
Beijing for the month of August during 2005 - 2016.




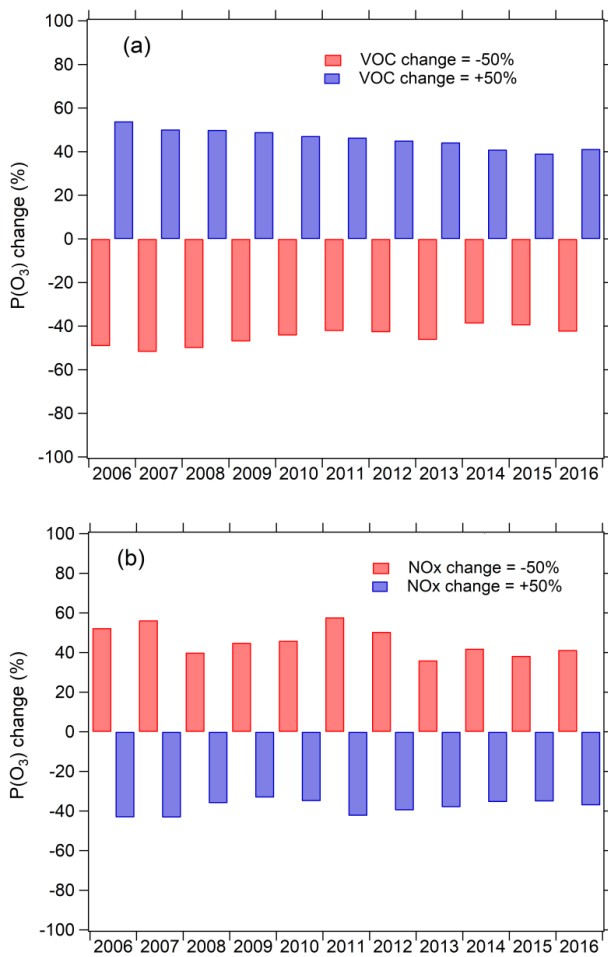


Figure 10. Sensitivity of monthly daytime mean P(O₃) to VOCs and NOx simulated
by box model during 2006 - 2016. VOCs and NOx is increased by 50% or decreased
by 50% to test the fractional change of monthly daytime mean P(O₃).








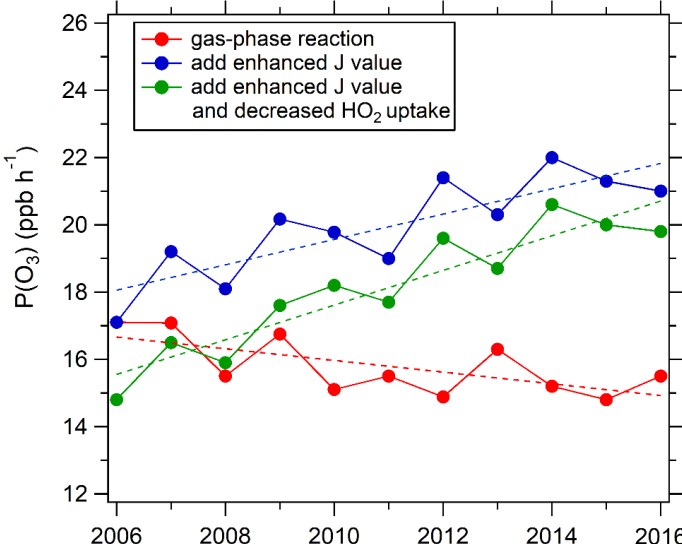


Figure 11. Trend of monthly daytime mean $P(O_3)$ simulated by the chemical box
model. Red dots: Only the gas-phase reactions are considered in the box model
constrained by observed photolysis frequencies from 2006 for all eleven years. Blue
dots: the box model as above, but constrained by the photolysis frequencies derived
for each year. Green dots: the box model constrained by the photolysis frequencies
derived for each year with the changing aerosol uptake of $HO_2$ also considered.







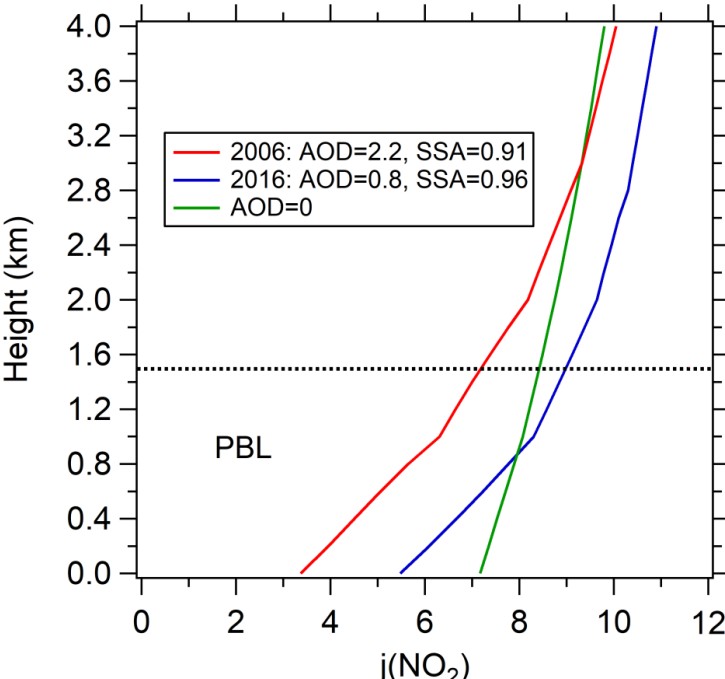


Figure 12. Vertical profiles of j(NO₂) simulated by the TUV model in Beijing. Three
scenarios are simulated: The model parameters are: (1) AOD=2.2, SSA=0.91 in August
2006; (2) AOD=0.8, SSA=0.96 in August 2016; (3) AOD=0. The daytime average
SZA=53°is used for all simulations. Dotted line represent the top of boundary layer.





Table 1. Instruments deployed in the measurement undertaken in August during 2005 -
2016 and used for data analysis.

| Parameters | Measurement technique | Time resolution | Detection limit | Accuracy |
|---|---|---|---|---|
| Photolysis frequencies | Spectroradiometer | 10 s | / | ± 10% |
| $O_3$ | UV photometry | 60 s | 0.5 ppbv | ± 5% |
| NO | Chemiluminescence | 60 s | 60 pptv | ± 20% |
| $NO_2$ | Chemiluminescence | 60 s | 300 pptv | ± 20% |
| CO | IR photometry | 60 s | 4 ppb | ± 5% |
| $SO_2$ | Pulsed UV fluorescence | 60 s | 0.1 ppbv | ± 5% |
| HCHO | Hantzsch fluorimetry | 60 s | 25 pptv | ± 5% |
| C2-C10VOCs | GC-FID/MS | 1 h | 20-300 pptv | ± 15~20% |
| $PM_{2.5}$ | TH-2000 | 60s | 1µg m$^{-3}$ | ± 5% |
| $S_a$ | SMPS | 60s | / | ±3% |
| AOD, SSA, AE | CIMEL Sun photometer | 5min | 0.01 | ±5% |










Table 2. Description of Ozone Metrics used in this study.

| categories | metric | definition |
| --- | --- | --- |
| general level | median (ppb) | 50th percentile of hourly concentrations |
| | MDA8 (ppb) | daily maximum 8 h average; the mean MDA8 O$_3$ in August of each year is used in this study. |
| | DTAvg (ppb) | daytime average ozone is the average of hourly ozone concentrations for the 12 h period from 07:00 to 19:00 local time |
| extreme level | MDA1 (ppb) | daily maximum 1 h average; the mean MDA1 O$_3$ in August of each year is used in this study. |
| | Perc98 (ppb) | 98th percentile of hourly concentrations |
| | 4MDA8 (ppb) | 4th highest MDA8 |
| ozone exposure | AOT40 (ppb h) | cumulative hourly ozone concentrations of >40 ppb |
| | SOMO35 (ppb day) | sum of positive differences between MDA8 and a cutoff concentration of 35 ppb |
| Exceedance days | NDGT70 (day) | total number of days with MDA8 values of >70 ppb |
| | Exceedance (day) | number of days with the ozone concentration exceeding the Chinese grade II national air quality standard, defined as MDA8 > 160 μg m$^{-3}$ |





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
