# Peer review of "Exploring the drivers of the increased ozone production in Beijing in"

_Atmospheric Chemistry and Physics, 2020_

## Referee Comment (RC1) · Ke Li (Referee) · 12 Jul 2020

Wang et al. examined the role of changes in precursor emissions and PM2.5 levels in recent ozone increase in summertime Beijing. By applying box modeling constrained through extensive measurement data in Beijing, they find that PM2.5 decrease is a major driver for the past- decade ozone increase despite the reduction of NOx and VOC emissions. Supported by fast increase of J(NO2), they show that the enhanced actinic flux by decrease PM2.5 is a more important factor than heterogeneous chemistry for ozone increase.

The topic of this work is well within the scope of ACP. Recent modeling studies have demonstrated the ozone-PM2.5 linkage to explain recent ozone increase in China that is of great concern for scientific communities. This study, from an observational perspective, admirably attempted to resolve the ozone increase by taking advantage of the valuable long-term measurements in Beijing. Overall, this manuscript is well structured and easily accessible. It will deepen our understanding of ozone-PM2.5 linkage, and in particular stimulate further studies to reconcile the discrepancy between modeling and observation-based studies. I think it is publishable in ACP after my following concerns are addressed.

1. There is gap between ozone production and its concentration. A recent ACPD paper (Gao J.H. et al., doi:10.5194/acp-2020-140) and also references in the second paragraph of their Introduction Section highlighted that decreased ozone production by PM2.5 via affecting photolysis rates is much more than the reduction in surface ozone concentration. Moreover, a lot of different transport model studies (Xing J. et al., doi:10.5194/acp-17-9869-2017; Li J. et al. doi: 10.1016/j.scitotenv.2017.12.041; Li K. et al., doi:10.1038/s41561-019-0464-x) also show that the impact of PM2.5 on summer surface ozone is not important. I suggest, at least, the authors to do some detailed discussion to reconcile this important issue. This is particularly helpful for future studies.

2. Diurnal variation of ozone production. The authors take daytime average over 7:00-19:00 or (6:00-18:00?) for ozone production. I am not sure if the results may differ by narrowing the average to afternoon hours when ozone production is active and HOx levels are high. Also, Hollaway et al. (doi:10.5194/acp-19-9699-2019) show that PM2.5 impacts on the summertime photolysis of NO2 and ozone level at surface in Beijing are important before 11 am and after 3 pm but very limited in afternoon hours. I suggest the authors to show some diurnal information of simulated ozone production.

3. The authors show an important result of an increased SSA in Beijing (Fig.12). More importantly, there is a shift pattern of j(NO2) over 2006-2016 that the crossing point between J(NO2) profile of 2006 and zero AOD profile changed from above PBL to below PBL in 2016. I think this means that the role of PM2.5 may be more important under condition like 2006, but will be limited under condition like 2016 when there is offsetting effect for PBL ozone by vertical mixing. This may deserve a discussion.

4. Some other specific comments. (1) It is confused to see 2005-2016 and 2006-2016 in the text. Please clarify this. (2) I suggest to use p-value other than the r square where there is a trend analysis. (3) Line 290: Shanghai should be "the south to North China Plain". (4) Lines 293-296: how about the role of regional contribution outside of Beijing? For example, the increasing emissions in the whole North China Plain. (5) Line 494: Please take caution when saying "ozone increase". you mean surface ozone concentrations?

---

## Referee Comment (RC2) · Anonymous Referee #2 · 30 Aug 2020

This manuscript investigated the trends of surface ozone metrics and their causes at a urban site in Beijing. Detailed measurements, TUV and RACM box model were employed. Long-term ozone trends in China is a key issue to take effective measures in ths situation of rapid increasing ozone backgroud. This manuscript investimated the impacts of meteorology, VOCs, NOx and PM2.5 on ozone trend, and would be helpful to understand the formation mechnisms in North China. This manuscript is within the scope of ACP and is well organised and executed and there is no doubt about the quality of the work. It can be accepted for publication after the following comments are addressed. 1. Regional transport is also a key source of surface ozone. This work tried to assess the impact of regional O3 by analyzing measurements at a

regional background site. This is not suffcient bacause this site was largely affected by Beijing emissions. I suggested that other background sites can be employed or back-transjectories at typical year can be used to analyze the impact of regional transport.

2. Recent, a few heteorogeneous chemical reactions are thought to be potential factors of ozone. For example, photolysis of HNO3 (NO3−) adsorbed on the solid surface of aerosol particles effectively produces HONO and NOx in the gas phase (Salgado and Rossi et al., PCCP,2002; Ramazan, 2006). A short disscuss should be performed.

M. S. Salgado Muñoz and M. J. Rossi, Heterogeneous reactions of HNO3 with flame soot generated under different combustion conditions. Reaction mechanism and ki-netics, Physical Chemistry Chemical Physics, Phys. Chem. Chem. Phys., 2002,4, 5110-5118 .

Ramazan, K., Wingen, A.M., Miller, Y., Chaban, G.M., Gerber, R.B., Xantheas, S.S. and Finlayson-Pitts, B.J. (2006). New experimental and theoretical approach to the heterogeneous hydrolysis of NO2: ̇ Key role of molecular nitric acid and its com-plexes. J. Phys. Chem. A 110: 6886–6897

---

## Author Comment (AC1) · 5 Oct 2020

Author's response by Wenjie Wang et al. Corresponding to li_xin@pku.edu.cn. We greatly appreciate the time and efforts that the Referees spent in reviewing our manuscript. The comments are really thoughtful and helpful to improve the quality of our paper. We have addressed each comment below, with the Referee comment in black text, our response in blue text, and relevant manuscript changes noted in red text.

Please also note the supplement to this comment:
https://acp.copernicus.org/preprints/acp-2020-434/acp-2020-434-AC1-supplement.zip

---

## Author Response (AR1)

Corresponding to li\_xin@pku.edu.cn.

We greatly appreciate the time and efforts that the Referees spent in reviewing our manuscript. The comments are really thoughtful and helpful to improve the quality of our paper. We have addressed each comment below, with the Referee comment in black text, our response in blue text, and relevant manuscript changes noted in red text.

\_\_\_\_\_

**Referee #1**

1. There is gap between ozone production and its concentration. A recent ACPD paper (Gao J.H. et al., doi:10.5194/acp-2020-140) and also references in the second paragraph of their Introduction Section highlighted that decreased ozone production by PM2.5 via affecting photolysis rates is much more than the reduction in surface ozone concentration. Moreover, a lot of different transport model studies (Xing J. et al., doi:10.5194/acp-17-9869-2017; Li J. et al. doi: 10.1016/j.scitotenv.2017.12.041; Li K. et al., doi:10.1038/s41561-019-0464-x) also show that the impact of PM2.5 on summer surface ozone is not important. I suggest, at least, the authors to do some detailed discussion to reconcile this important issue. This is particularly helpful for future studies.

Response: I agree with you that there is gap between ozone production and its concentration. I have given some detailed discussion to reconcile this important issue. Line 542 - 558: Several three-dimension transport model studies show that the impact of PM2.5 on summer surface ozone is not important (Li et al., 2018;Li et al., 2019c;Li et al., 2019b;Xing et al., 2017). Moreover, a recent study highlighted that decreased ozone production by PM2.5 via affecting photolysis rates is much more than the reduction in surface ozone concentration (Gao et al., 2020). The difference between the two reductions in ozone production and surface ozone concentration indicates that, in addition to ozone photochemistry, there must be other ozone related physical processes influenced by the reduction in photolysis rate induced by aerosols. Model simulation indicates that aerosols lead to high concentrations of ozone aloft being

entrained by turbulence from the top of the planetary boundary layer (PBL) to the surface by altering photolysis rate and partly counteracting the reduction in surface ozone photochemical production induced by aerosols. In addition, the impact of aerosols on ozone from local and adjacent regions was more significant than that from long-distance regions (Gao et al., 2020). The accurate quantification of the effects of vertical mixing and long-distance transport on surface ozone concentration plays a critical role in the impact of aerosols on surface ozone, which needs further study in the future.

2. Diurnal variation of ozone production. The authors take daytime average over 7:00-19:00 or (6:00-18:00?) for ozone production. I am not sure if the results may differ by narrowing the average to afternoon hours when ozone production is active and HOx levels are high. Also, Hollaway et al. (doi:10.5194/acp-19-9699-2019) show that PM2.5 impacts on the summertime photolysis of NO2 and ozone level at surface in Beijing are important before 11 am and after 3 pm but very limited in afternoon hours. I suggest the authors to show some diurnal information of simulated ozone production.

Response: I have analyzed the trend of simulated  $P(O_3)$  in the afternoon hour (12:00-15:00), which increased at a rate of 1.3% yr-1, lower than the increasing rate of daytime average  $P(O_3)$ . Diurnal variation of simulated  $P(O_3)$  in 2013 is shown in Figure S1. The diurnal variation of simulated  $P(O_3)$  in this study indicates that the influence of aerosols on  $P(O_3)$  is still significant in the afternoon leading to  $P(O_3)$  decreased by ~17%, which is slightly lower than the decrease in the whole daytime (25%) (Figure S2). This is because that the average AOD in the afternoon (1.4) is significantly higher than that before 11:00 am (0.94) and after 3:00 pm (1.1) despite lower SZA and lower light absorptive ability (i.e. higher SSA) in the afternoon. Line 461-472: The simulated  $P(O_3)$  in the afternoon hour (12:00-15:00) when ozone production is active and HOx levels are high increased at a rate of 1.3% yr-1, which is lower than the increasing rate of daytime average  $P(O_3)$  (2.2% yr-1) (Figure S2). Hollaway et al. (2019) show that the impacts of aerosols on the summertime

photolysis of NO2 and ozone at surface in Beijing are important before 11:00 am and after 3:00 pm but very limited in afternoon hours due to lower SZA and lower light absorptive ability of aerosol in the afternoon. However, the diurnal variation of simulated  $P(O_3)$  in this study indicates that the influence of aerosols on  $P(O_3)$  is still significant in the afternoon leading to  $P(O_3)$  decreased by ~17%, which is slightly lower than the decrease in the whole daytime (25%) (Figure S3). This is because that the average AOD in the afternoon (1.4) is significantly higher than that before 11:00 am (0.94) and after 3:00 pm (1.1) despite lower SZA and lower light absorptive ability (i.e. higher SSA) in the afternoon.

Figure S2. Trend of monthly afternoon (12:00-15:00) mean P(O3) simulated by the chemical box model. Red dots: Only the gas-phase reactions are considered in the box model constrained by observed photolysis frequencies from 2006 for all eleven years. Blue dots: the box model as above, but constrained by the photolysis frequencies derived for each year.

Figure S3. Diurnal variation of simulated  $P(O_3)$  in Beijing in August during 2005-2016.  $P(O_3)_{j_obs}$  represents ozone production rate under observed photolysis frequencies;  $P(O_3)_{j_AOD=0}$  represents ozone production rate under calculated photolysis frequencies when AOD is equal to 0.

3. The authors show an important result of an increased SSA in Beijing (Fig.12). More importantly, there is a shift pattern of j(NO2) over 2006-2016 that the crossing point between J(NO2) profile of 2006 and zero AOD profile changed from above PBL to below PBL in 2016. I think this means that the role of PM2.5 may be more important under condition like 2006, but will be limited under condition like 2016 when there is offsetting effect for PBL ozone by vertical mixing. This may deserve a discussion.

Response: Yes, I agree with you. This is a good point. I have given a brief discussion according to your suggestion.

Line 533-541: However, there is a shift in the vertical profile of  $j(NO_2)$  that is important. The crossing point between  $j(NO_2)$  profile of 2006 and zero AOD profile is below PBL, while in 2016 the  $j(NO_2)$  profile crosses the zero AOD profile within the PBL. This means that as the AOD is reduced further, changes in the vertical average j(NO2) will be limited, since increases in j(NO2) near the top of the PBL will compensate for decreases near the surface. Additionally, this also denotes that the role of PM2.5 may be more important under condition like 2006, but will be limited under condition like 2016 when there is offsetting effect for PBL ozone by vertical mixing caused by larger ozone vertical gradient (Gao et al., 2020).

4. Some other specific comments. (1) It is confused to see 2005-2016 and 2006-2016 in the text. Please clarify this. (2) I suggest to use p-value other than the r square where there is a trend analysis. (3) Line 290: Shanghai should be "the south to North China Plain". (4) Lines 293-296: how about the role of regional contribution outside of Beijing? For example, the increasing emissions in the whole North China Plain. (5) Line 494: Please take caution when saying "ozone increase". you mean surface ozone concentrations?

Response: (1) NOx data in 2005 were not available. Therefore, the trend of NOx during 2006-2016 was analyzed. In addition, we focus on the trend of P(O3) during the period of 2006-2016 due to the lack of NOx data in 2005. (2) Thank you. I have summarized p-values for the temporal trend of all parameters in table 2. (3) Thank you. I have revised it. (4) In this study, we mainly focus on the variation of the local ozone production. It is difficult to give an accurate estimation of the regional contribution outside of Beijing. Previous studies have reported that regional transport from neighboring provinces outside Beijing (including Hebei, Tianjin and Shandong) contributed about 35%-60% of ozone in Beijing during high ozone episodes (Streets et al., 2007; Wang et al., 2020). I have analyzed the ozone trend in Changdao site, a background site in the east of North China Plain, to discuss the regional contribution outside of Beijing due to increasing emissions in the whole North China Plain. This site is nearly not influenced by local anthropogenic emissions. MDA8 ozone concentrations at the Changdao site increased slowly (+1.2 ppbv yr-1,  $r^2 = 011$ , p=0.25) during 2013-2019, which is about a half of the increasing rate of MDA8 ozone concentrations at PKUER site (+2.3  $\pm$  1.2 ppbv yr-1, r2=0.66) during 2006-2016.

345-346: NOx data in 2005 were not available. Therefore, the trend of NOx during 2006-2016 was analyzed.

438-439: We focus on the period during 2006-2016 due to the lack of NOx data in 2005.

| Parameter                   | Period    | $r^2$ | p value | P value | P value |
|-----------------------------|-----------|-------|---------|---------|---------|
|                             |           |       |         | < 0.01? | < 0.05? |
| median                      | 2005-2016 | 0.63  | 0.002   | yes     | yes     |
| perc98                      | 2005-2016 | 0.11  | 0.288   | no      | no      |
| DTAvg                       | 2005-2016 | 0.47  | 0.014   | no      | yes     |
| MDA1                        | 2005-2016 | 0.32  | 0.057   | no      | no      |
| MDA8                        | 2005-2016 | 0.66  | 0.001   | yes     | yes     |
| 4MDA8                       | 2005-2016 | 0.42  | 0.023   | no      | yes     |
| AOT40                       | 2005-2016 | 0.67  | 0.001   | yes     | yes     |
| NDGT70                      | 2005-2016 | 0.56  | 0.005   | yes     | yes     |
| SOMO35                      | 2005-2016 | 0.57  | 0.004   | yes     | yes     |
| exceedance                  | 2005-2016 | 0.32  | 0.054   | no      | no      |
| Ox                          | 2005-2016 | 0.38  | 0.044   | no      | yes     |
| CO                          | 2005-2016 | 0.87  | 0.001   | yes     | yes     |
| VOC reactivity              | 2005-2016 | 0.52  | 0.006   | yes     | yes     |
| NOx                         | 2006-2016 | 0.81  | 0.001   | yes     | yes     |
| Calculated                  | 2006-2016 | 0.94  | 0.000   | yes     | yes     |
| j(NO 2 )         |           |       | 0.000   |         |         |
| AOD (380 nm)                | 2006-2016 | 0.78  | 0.000   | yes     | yes     |
| PM 2.5           | 2009-2016 | 0.93  | 0.000   | yes     | yes     |
| Sa                          | 2006-2016 | 0.51  | 0.010   | yes     | yes     |
| SSA                         | 2005-2016 | 0.70  | 0.001   | yes     | yes     |
| AE                          | 2005-2016 | 0.03  | 0.593   | no      | no      |
| COT                         | 2005-2016 | 0.003 | 0.875   | no      | no      |
| Total O 3 column | 2005-2016 | 0.15  | 0.215   | no      | no      |

Line 657-658:

Table 2. p value of temporal trends for different parameters.

Line 307-309: Additionally, there were very small trends of  $O_3$  concentrations at the background site (Dongtan) in Shanghai, located to the south of the North China Plain (Gao et al., 2017).

Line 309-317: However, these background sites in Beijing and Shanghai may be strongly affected by local emissions. MDA8 ozone concentrations at the Changdao site, a background site in the east of the North China Plain that is much less influenced by local emissions, increased slowly (+1.2 ppbv yr-1, r2=0.11), but that rate is not statistically significant (p = 0.25) during 2013-2019 (Figure S1). Based on these reports of smaller and variable trends, we assume that the trend in regional background ozone in the North China Plain made only a minor contribution to the relatively larger ozone trend observed at the PKUERS site (+2.3  $\pm$  1.2 ppbv yr-1, r2=0.66, p = 0.001).

Figure S1. The trend of average MDA8 ozone in Changdao during 2013-2019. These data is acquired from "Blue book on prevention and control of atmospheric ozone pollution in China (in Chinese)" reported by Chinese Society of Environmental Sciences in 2020 (http://www.epserve.com/forepart/zxnr\_index.do?oid=51478637&tid=26378242).

**Referee #2**

1. Regional transport is also a key source of surface ozone. This work tried to assess the impact of regional O3 by analyzing measurements at a regional background site. This is not sufficient because this site was largely affected by Beijing emissions. I suggested that other background sites can be employed or backtransjectories at typical year can be used to analyze the impact of regional transport.

Response: I agree with you that the regional background site in Beijing was largely affected by Beijing emissions. I have chosen another site in Changdao, Shandong province, which is nearly not influenced by local emissions and thus is a better background site in North China Plain. MDA8 ozone concentration at the Changdao site increased at a rate of 1.2 ppbv yr-1 ( $r^2$ =0.11) during 2013-2019, which is significantly smaller than that at PKUER site during 2006-2016 (2.3 ppbv yr-1,  $r^2$ =0.66) and during 2013-2019 (2.0 ppbv yr-1,  $r^2$ =0.67).

Line 309-317: However, these background sites in Beijing and Shanghai may be stro

---

## Author Response (AR2)

Corresponding to  li_xin@pku.edu.cn.

We greatly appreciate the time and efforts that the Referees spent in reviewing our manuscript. The comments are really thoughtful and helpful to improve the quality of our paper. We have addressed each comment below, with the Referee comment in black text, our response in blue text, and relevant manuscript changes noted in red text.

1. Lines 541-558. This part is inserted upon comment/revision but is a bit difficult to read. In this part ozone "DECREASE" is described as sensitivities with aerosol "INCREASES", while in the previous part ozone change (mostly increase) is described with aerosol "TEMPORAL" change (i.e., aerosol decrease). Although most of the sentences here were taken directly from Abstract of Gao et al. (2020), I would suggest rewriting them in the context of this manuscript, to avoid any confusion or misunderstanding.

Response: Many thanks, I have revised it according to your suggestions.

Line 542-561:

Quantitative studies suggested that, the impact of aerosols via affecting photolysis rates on net ozone production (Cai, 2013;Wang et al., 2019;Castro et al., 2001) is more than on surface ozone concentrations (Jacobson, 1998;Li et al., 2011a;Li et al., 2011b;Wang et al., 2016a). Moreover, several different transport model studies (Xing et al., 2017;Li et al., 2018a;Li et al., 2019c) also show that the impact of $PM_{2.5}$ on summer surface ozone concentrations is not important, which is different from the relatively large change of net ozone production in this study. There are several possible causes of this difference: (1) The net ozone production characterizes the local ozone production level but can't completely represent ozone concentrations that are also influenced by regional transport of ozone (Streets et al., 2007;Wang et al., 2020;Moghani et al., 2018). (2) The light-extinction effect of aerosols lead to ozone at the top of PBL being entrained by turbulence to the surface to partly counteract the reduction in surface ozone photochemical production induced by aerosols (Gao et al., 2020). (3) AOD decreased by 58% in Beijing in this study, which is higher than that of regional AOD decrease in eastern China (20%) (Li et al.,

2019a). (4) The photolysis rates in this study are determined using the observed AOD SSA, and confirmed by ground-level measurements of photolysis frequencies, while photolysis rates are determined using online algorithm in other transport model studies. The different calculation procedures also contribute to the different results between our study and other transport model studies.

2. Line 533. above PBL (not below PBL)?
Response: Many thanks, I have revised it.
Line: 533-535:
The crossing point between $j(NO_2)$ profile of 2006 and zero AOD profile is above PBL, while in 2016 the $j(NO_2)$ profile crosses the zero AOD profile within the PBL.

3. Line 567 (previous Line 494): Please respond to the comment from the referee #1: Please take caution when saying "ozone increase". you mean surface ozone concentrations?
Response: Many thanks, I have revised it.
Line 567-570:
Consistent with the implementation of stringent emission control measures, concentrations of $PM_{2.5}$ and ozone precursors (VOCs and NOx) decreased rapidly, but in contrast surface $O_3$ and Ox concentrations increased.

4. Line 278. Remove "was"
Response: Many thanks, I have revised it.

[revised manuscript text omitted]